# Effects of subclinical depression on prefrontal–striatal model-based and model-free learning

Suyeon Heo[1,2], Yoondo Sung[1], Sang Wan Lee[1,2,3,4,5] *

**1** Department of Bio and Brain Engineering, Korea Advanced Institute of Science and Technology (KAIST), Daejeon, Republic of Korea, **2** Brain and Cognitive Engineering Program, Korea Advanced Institute of Science and Technology (KAIST), Daejeon, Republic of Korea, **3** KAIST Institute for Health Science Technology, Korea Advanced Institute of Science and Technology (KAIST), Daejeon, Republic of Korea, **4** KAIST Institute for Artificial Intelligence, Korea Advanced Institute of Science and Technology (KAIST), Daejeon, Republic of Korea, **5** KAIST Center for Neuroscience-inspired AI, Korea Advanced Institute of Science and Technology (KAIST), Daejeon, Republic of Korea

* sangwan@kaist.ac.kr

**Data Availability Statement:** The raw behavioral data, simulation codes, and fMRI results are available for download at https://github.com/brain-machine-intelligence/Depression-study. Numerical

## Abstract

Depression is characterized by deficits in the reinforcement learning (RL) process. Although many computational and neural studies have extended our knowledge of the impact of depression on RL, most focus on habitual control (model-free RL), yielding a relatively poor understanding of goal-directed control (model-based RL) and arbitration control to find a balance between the two. We investigated the effects of subclinical depression on model-based and model-free learning in the prefrontal–striatal circuitry. First, we found that subclinical depression is associated with the attenuated state and reward prediction error representation in the insula and caudate. Critically, we found that it accompanies the disrupted arbitration control between model-based and model-free learning in the predominantly inferior lateral prefrontal cortex and frontopolar cortex. We also found that depression undermines the ability to exploit viable options, called exploitation sensitivity. These findings characterize how subclinical depression influences different levels of the decision-making hierarchy, advancing previous conflicting views that depression simply influences either habitual or goal-directed control. Our study creates possibilities for various clinical applications, such as early diagnosis and behavioral therapy design.

## Author summary

Human decision making is known to be driven by at least two distinct processes, goal-directed and habitual learning. Previous studies argued that these systems and their interaction are disrupted in depression. However, we have limited understanding of the integration of the two systems and how this case extends to early or mild depression. We used a computational model and fMRI to address this issue. We found that depression-related changes were observed in the different levels of the decision making process. Notably, we found that depressive individuals have higher sensitivity of the habitual learning process,

data related to figures is within its Supporting Information files.

**Funding:** This research was supported by the National Research Foundation of Korea (NRF) grant funded by the Korea government (MSIT) (NRF-2019M3E5D2A01066267) (SWL and YS), Institute for Information & Communications Technology Promotion (IITP) grant funded by the Korea government (MSIT) (No.2019-0-01371, Development of brain-inspired AI with human-like intelligence) (SWL and YS) and the Brain Research Program through the National Research Foundation of Korea (NRF) funded by the Ministry of Science, ICT & Future Planning (NRF-2016M3C7A1914448) (SWL and SH). The funders had no role in study design, data collection and analysis, decision to publish, or preparation of the manuscript.

**Competing interests:** The authors have declared that no competing interests exist.

indicating the impairment of the proper integration of the two. Our findings raise the hope about developing clinical applications for the early diagnosis of this disorder, as well as using behavioral/cognitive therapy or brain stimulus techniques for restoring the balance between goal-directed and habitual learning in individuals with subclinical depression.

## Introduction

Major depressive disorder (MDD) has received considerable attention, as the lifetime prevalence of the disorder is higher than 10% worldwide [1]. MDD is characterized by deficits in decision-making [2,3] and its underlying reward learning processes [4]. Recently, with the development of computational models, several studies have explored how depression influences the reward learning system.

Reinforcement learning (RL), the process of learning to develop a behavioral policy to maximize reward [5], has been known to be guided by the two distinct RL strategies: model-based (MB) RL and model-free (MF) RL, each of which guides goal-directed and habitual RL, respectively [6,7]. Model-based RL guides context-sensitive and goal-directed behaviors through a sophisticated process in which the learning agent makes decisions by simulating an internal environmental model, whereas model-free RL is associated with habitual responses to reward-predicting stimuli based on learned associations between stimuli and rewards [6,8]. Mounting evidence suggests that depression is characterized by impairments in RL. For example, behaviors in depressive individuals can be accounted for by impaired model-based RL [9–11] or a transition from model-based to model-free RL [12]. However, most studies have focused on the effect of depression on model-free RL. For instance, depressive people exhibit an impaired ability to learn stimulus–reward associations accompanying inaccurate representations of reward prediction error [13–17] or abnormal learning rate control [12,18].

Impairment in RL is associated with not only the onset of depression, but also the development of depression. For instance, stress, one of the major risk factors for depression [19,20], can induce deficits in RL. Previous findings have shown that people exhibit a reduced ability to engage in model-based RL under conditions of chronic [21,22] and acute [23,24] stress (For review, refer [25]). These findings suggest a gradual impairment of RL from the very early stages of depression.

Although these studies have contributed to our understanding of depression in the context of RL, it is still unclear whether depression is best characterized by model-free RL, model-based RL, or an interaction between the two, or how depression influences the neural circuits guiding goal-directed and habitual behavioral control. Moreover, little is known about how these cases extend to early or mild depression.

Here, we aim to provide a computational and neural account of how subclinical depression affects goal-directed and habitual control in the prefrontal–striatal circuitry. First, we ran a model comparison analysis to identify a version of the RL model that best explains the choice patterns of human subjects. The purpose of this analysis is to investigate the parametric effects of subclinical depression on model-based and model-free RL, which would lead to the discovery of novel behavioral traits of depression. In particular, our computational models consider the sub-optimality of RL, allowing us to explain choice behavior patterns across a wide spectrum of depression. We combined this with model-based functional magnetic resonance imaging (fMRI) to identify the parametric effects of subclinical depression on neural systems associated with model-based and model-free RL. In the subsequent analysis, we attempted to

fully characterize how subclinical depression disrupts the arbitration between model-based and model-free RL by combining the results from the computational modeling and model-based fMRI analyses and the multi-voxel pattern analysis. Note this is the first attempt to establish a link between depressive symptom and neural circuits underlying model-based and model-free RL.

## Results

### Detrimental effects of subclinical depression on task performance

63 participants conducted a sequential two-stage Markov decision task (Fig 1, [26]). We ran model-free behavioral analyses to explore pure behavioral effects of subclinical depression on choice behavior. For this, we used three behavioral measures, each of which is focused on evaluating overall task performance, performance associated with model-based learning, and performance associated with model-free learning. Overall task performance is negatively correlated with the self-reported depression score. The accumulated reward throughout the task decreases significantly as individual depression score (CES-D) increases (correlation coefficient estimate = -0.584 [$p$ = 4.94e-07]; Fig 2A). The choice optimality, the measure that

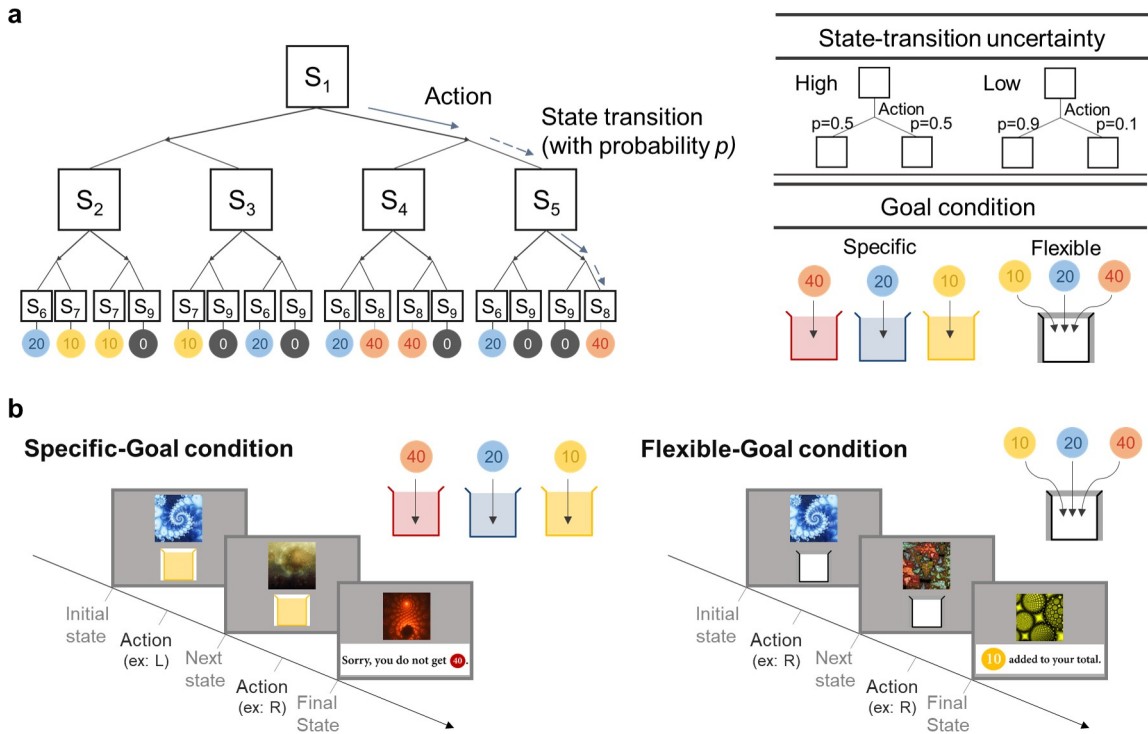

**Fig 1. Markov decision task structure.** We use the two-stage Markov decision task proposed by Lee et al. (2014). (a) Two-stage Markov decision task. In each stage, participants make a binary choice (left or right). After the first choice in the initial state ($S_1$), they were moved forward to one of four states in the second stage ($S_2$ or $S_3$ when making the 'left' choice, $S_4$ or $S_5$ when making the 'right' choice) with certain state-action-state transition probability $p$. The transition probability (0.5, 0.5) and (0.9, 0.1) corresponds to a high-uncertainty and a low-uncertainty environment, respectively. The task consists of the two goal conditions: a specific-goal and a flexible-goal condition. In the specific goal condition, subjects can collect coins (redeemable for monetary reward) only if the coin color matches with the color of the token box (red, blue, yellow). In the flexible goal condition indicated by the white token box, all types of coins are redeemable. (b) Illustration of the task. Each state is associated with a specific fractal image. When the initial state is shown on the screen, participants make a binary choice, and by doing so they proceed to the next state; the state transition follows the state-transition probability. In the next state, the participants make a binary choice to get to an outcome state. Each outcome state is associated with a specific coin (red, blue, yellow). In the specific-goal condition, participants collect the coin only when the color of the coin matches with the color of the given box, whereas in the flexible-goal condition, they collect coins regardless of its color.

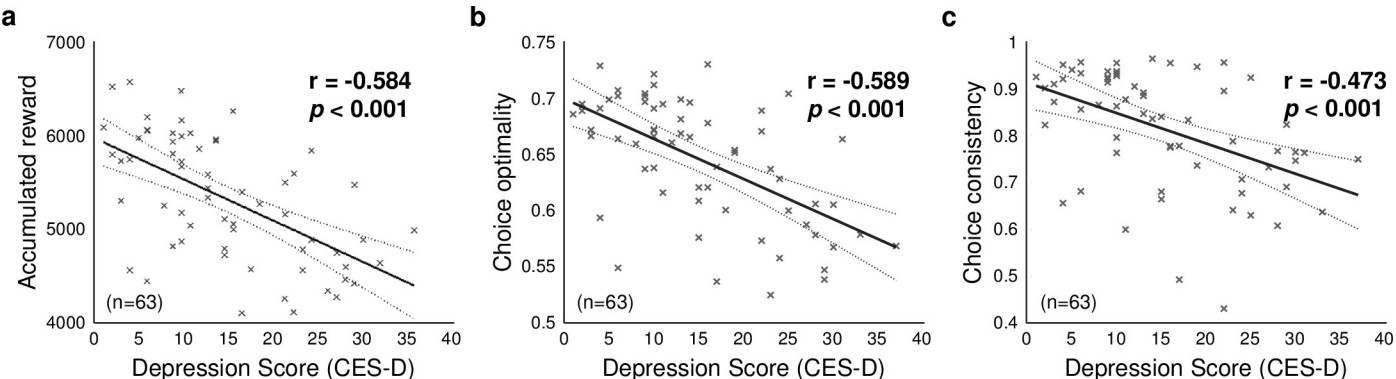

**Fig 2. Behavioral results.** (a) Relationship between the individual depression score (CES-D) and accumulated reward (n = 63). The task performance decreases as the depression score increases. (b) Relationship between depression score and the choice optimality (n = 63). The choice optimality is inversely proportional to the depression score. (c) Relationship between the depression score and choice consistency in the first state ($S_1$) (n = 63). The choice consistency index is negatively correlated with the depression score.

quantifies the extent to which a subject's choice reflects an optimal policy, is also inversely proportional to the CES-D score (correlation coefficient estimate = -0.589 [$p$ = 3.78e-07]; Fig 2B). Finally, choice consistency, the proportion of making the same choice as in previous trials, decreases as the CES-D score increases (correlation coefficient estimate = -0.473 [$p$ = 8.92e-05]; Fig 2C). These results demonstrate that subclinical depression has a damaging effect on RL process, leading to suboptimal choices.

## Computational model to account for suboptimal arbitration control between model-based and model-free learning

We adopted the previous dynamic arbitration control hypothesis that respective prediction uncertainty—specifically, the amount of uncertainty in the state and reward prediction error of model-based and model-free RL—mediates the trial-by-trial value integration of the model-based and model-free systems [26]. To fully explore the effects of subclinical depression on arbitration control, however, a model should be flexible enough to account for any individual variability arising from suboptimal learning and decision-making.

Specifically, we hypothesized that prediction uncertainty mediates not only value integration, but also value–action conversion (Fig 3A). To this end, we redesigned the arbitration control scheme to allow for the sub-optimality of RL in both learning values and converting learned values into choice behavior. We examined the former by introducing separate learning rates for model-based and model-free RL and the latter by defining an exploitation sensitivity parameter as a function of model preference for either model-based or model-free RL. The first parameter setting, separate learning rates for the two RL systems, would allow us to extend the previous hypotheses concerning changes in learning rate of the MF system into that of the MB system [27]. The second parameter setting, the exploitation sensitivity parameter, would allow us to evaluate the hypothesis that final action selection is affected by the proportion of using the MB and MF system.

We compared prediction performance for the five different versions of arbitration control, including the original arbitration model and four other versions (separated learning rates, separated learning rates/$\tau$ = f($P_{MB}$); f: logistic, separated learning rates/$\tau$ = f($P_{MB}$); f: linear, separated learning rates/$\tau$ = f($\tau_{MB}$, $\tau_{MF}$, $P_{MB}$); f: weighted linear) implementing our hypothesis in different ways. We used a Bayes Factor (BF) [28] as a performance measure to compare the models. We found that the version implementing our hypothesis, in which the degree of

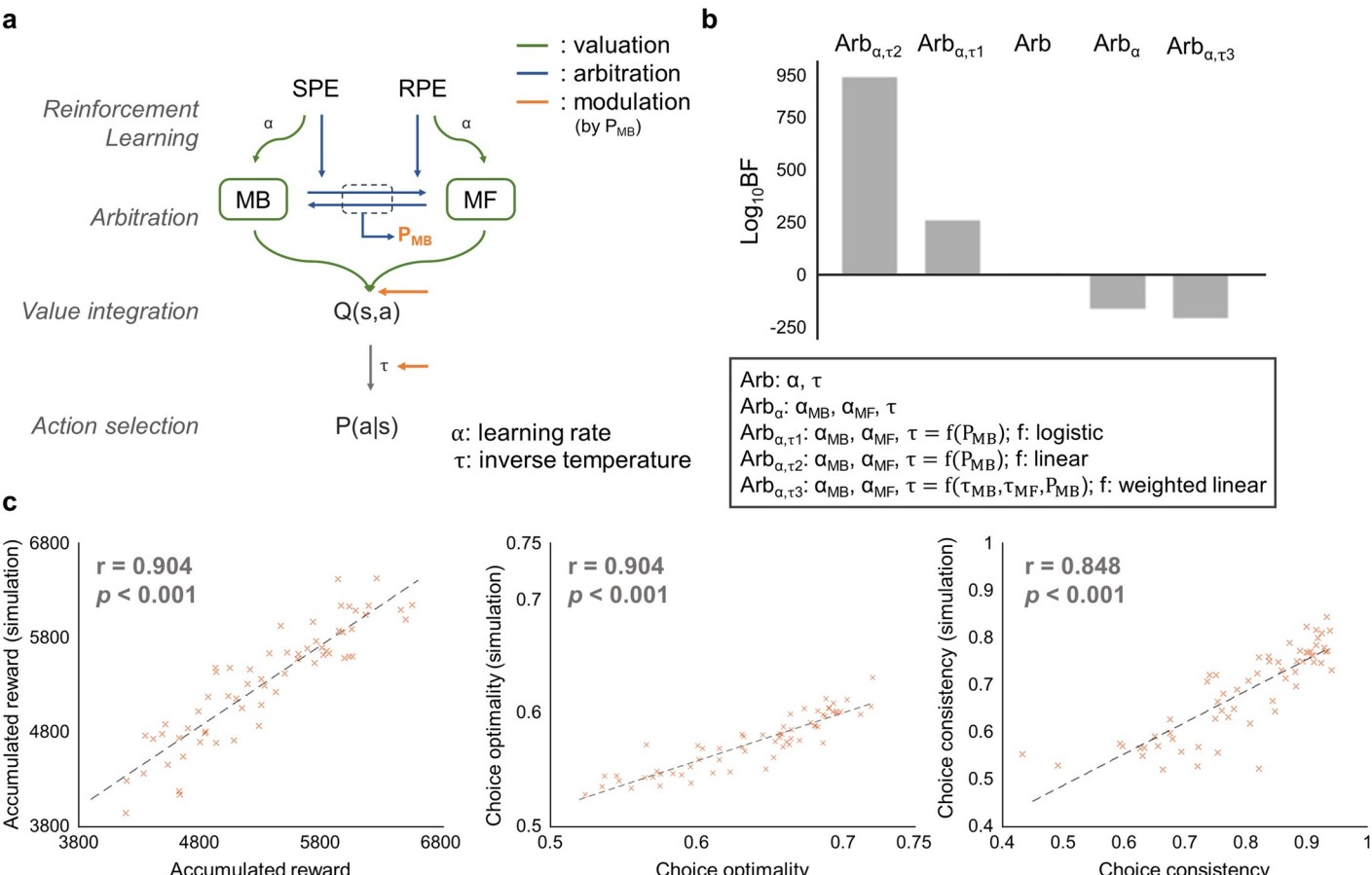

**Fig 3. Computational model of dynamic arbitration control.** (a) Computational model to investigate dynamic control mechanisms to arbitrate between model-based (MB) and model-free (MF) RL. Two separate RL systems update an action value and reliability of its prediction by using prediction errors (SPE and RPE, respectively; green). The reliability values of the two systems were then used to compute the model choice probability ($P_{MB}$) (blue). The model choice probability guides both the value integration and value-action conversion process; both effects are indicated by the orange arrow. (b) Model comparison analysis. We used a Bayes Factor (BF) for comparing the goodness of fit while penalizing for the model complexity. All BF are compared based on the previous model, Arb. Arb refers to the computational model proposed in (Lee et al., 2014). Additional versions of arbitration control consider separate learning rates and dynamic exploitation. $Arb_\alpha$ assigns separate learning rates to two different systems (i.e. use $\alpha_{MB}$, $\alpha_{MF}$ instead of $\alpha$). $Arb_{\alpha,\tau}$ is the same as $Arb_\alpha$, except for the assumption that the degree of exploitation is a function of the model choice probability. The best version of model uses the degree of exploitation ($\tau$) as a weighted sum of the model-based and the model-free exploitation parameter ($\tau_{MB}$ and $\tau_{MF}$, respectively) with the model choice probability $P_{MB}$ (i.e. $\tau = P_{MB}*\tau_{MB} + (1-P_{MB})*\tau_{MF}$). (c) Model recovery of the behavioral data. To assess whether the free parameters of the best version of the model encapsulate the essence of choice behavior, we ran the behavior recovery analysis. For this, we generated simulated behavioral data from the model fitted to each individual participant's original behavioral data. Shown are the comparison between behavioral measures from the original and the simulated behavioral data. Dots are the averaged simulated results for each subjects.

exploitation is determined by the weighted sum of the model-based and model-free exploitation parameters with the model choice probability (i.e. $\tau = P_{MB}*\tau_{MB}+P_{MF}*\tau_{MF}$, $P_{MF} = 1-P_{MB}$), explains the subjects' behavior (BF = $2.2\times10^{135}$, $t(62) = 2.46$, [$p = 0.017$]; BF and paired t-test comparison results with the second-best model, $Arb_\alpha$, respectively), significantly better than the original model (BF = $9.0\times10^{208}$) [26] (Fig 3B; for a model comparison, see Fig A in S1 File). To validate whether the best version of the model captures the underlying model of the human behavior, we generated the simulated behavioral data on each subject's estimated parameter (similar analysis as [29]). We performed 100 replications for each subject. The result shows that the performance of the simulated and the actual human is highly correlated (accumulated reward: r = 0.904, p<0.001; choice optimality: r = 0.904, p<0.001; choice consistency: r = 0.848, p<0.001; Fig 3C), supporting the reliability of the model. Our model comparison result not only corroborates the previous finding that prediction uncertainty mediates the arbitration

between model-based and model-free RL, but also demonstrates the effect of prediction uncertainty on both value integration and value–action conversion.

To check whether our computational model of arbitration captures the essence of subjects' behavior associated with depression symptom, we ran the post-hoc analyses in which we compared the subjects' data with our model's simulation data in terms of the three different behavioral measures: raw choice behavior, choice consistency [26], and choice optimality [30] (Fig 4). Note that choice consistency and choice optimality is known as a behavioral measure of model-free and model-based learning, respectively. To test the effect of depression on these behavior patterns, we split the subjects into "healthy group" (n = 15) and "subclinical depression group" (n = 13), based on the standard cutoff criterion of the CES-D score, 16 [31,32]. Note that this criterion was suggested to identify individuals at risk for clinical depression, not for diagnosing a depressive disorder. The results indicate that our model explains subjects' key behavioral patterns associated with model-based and model free learning, as well as the effect of depression on those behaviors.

## Neural evidence of model-based and model-free control

To further examine whether our model explains the neural activity patterns of brain areas previously implicated in model-based and model-free RL, we ran a model-based fMRI analysis in which each key signal of our computational model was regressed against the fMRI data. Note that we supplemented behavioral data after the fMRI experiment, so our fMRI data (n = 28) is a subset of the whole dataset used for behavioral analysis (n = 63).

First, we replicated previous findings concerning the neural correlates of prediction error for the model-based and model-free systems. The state prediction error (SPE) was found bilaterally in the insula and the dorsolateral prefrontal cortex (dlPFC) (all $p<0.05$ FWE corrected). The reward prediction error (RPE) was correlated with neural activity in both sides of the ventral striatum (both $p<0.05$ FWE corrected). These results are fully consistent with previous findings [26,33,34] (Fig 5 and Table A in S1 File).

We also successfully replicated previous findings supporting the neural hypothesis of arbitration control. We found the max reliability signal, the key signal used to mediate arbitration between model-based and model-free RL, in the bilateral inferior lateral PFC (ilPFC, left: $p<0.05$ in $p<0.05$ FWE corrected; right: $p<0.05$ in a 10 mm small-volume corrected [SVC] at [48,35,–2]) and the frontopolar cortex (FPC, $p<0.05$ FWE corrected), fully consistent with previous results [26,35] (Fig 5 and Table A in S1 File).

Next, we tested the brain areas implicated in value computation. The chosen value of the model-based system ($Q_{MB}$) was found to be encoded in the precentral gyrus ($p<0.05$ FWE corrected), a region known to be more involved in goal-directed choices, compared to non-goal-directed choices [36]. The chosen value of the MF system ($Q_{MF}$) was found in the dorsal ACC (dACC), premotor cortex, and dorsolateral PFC (all $p<0.05$ FWE corrected). The activation in the dACC is congruent with the fact the region is involved in the action-outcome association [37]. We also tested for the integrated value signal, expressed as a sum of the value estimates of the model-based and model-free systems weighted by the arbitration control signal ($P_{MB}$). The ventromedial PFC was positively correlated with the difference between the integrated value signals for the chosen and unchosen actions ($p<0.05$ FWE corrected), fully consistent with previous reports on choice values [38–40] (Fig 5 and Table B in S1 File).

Unlike the previous arbitration hypothesis [26], our computational model also predicted that arbitration control influences how integrated values are converted into actual choices. Finally, we attempted to identify the brain regions involved in value–action conversion. We found that the inferior parietal lobe, insula (all $p<0.05$ FWE corrected), middle frontal gyrus,

**a** **Comparison between human subjects' behavioral data and model simulation data**

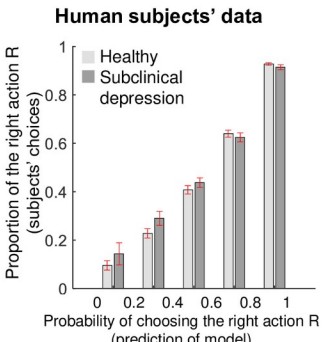

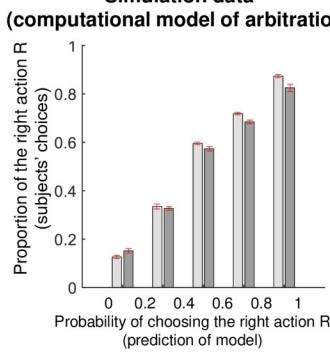

| Effect (repeated measure ANOVA) | p-value | |
| --- | --- | --- |
| | Subjects' data | Simulation data |
| Probability of choosing R | 3.75e-107 | 1.14e-158 |
| Depression score level x Probability of choosing R | 5.92e-02 | 1.34e-02 |

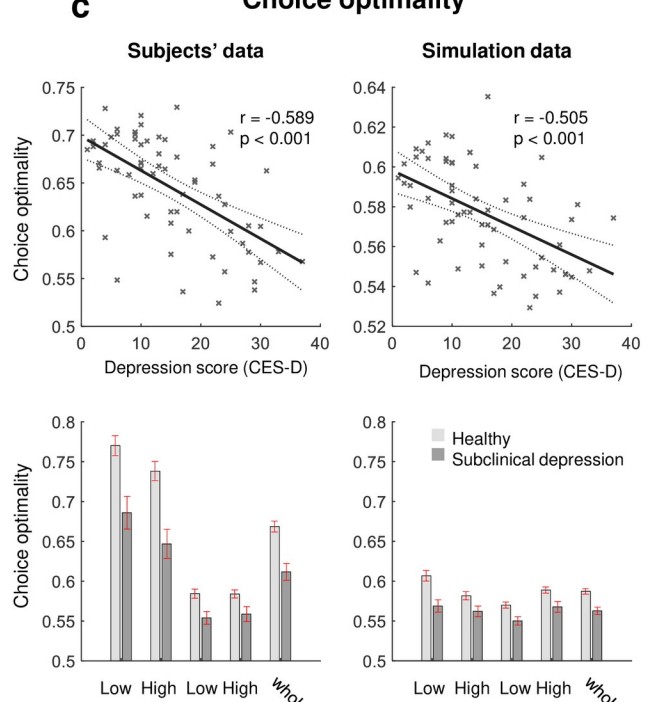

**b** **Choice consistency** **c** **Choice optimality**

| Effect (repeated measure ANOVA) | p-value | |
| --- | --- | --- |
| | Subjects' data | Simulation data |
| Goal | 8.69e-05 | 8.17e-16 |
| Uncertainty | 1.02e-02 | 1.93e-02 |
| Goal x uncertainty | 2.90e-01 | 2.96e-01 |
| Depression score level | 2.07e-04 | 1.27e-04 |

| Effect (repeated measure ANOVA) | p-value | |
| --- | --- | --- |
| | Subjects' data | Simulation data |
| Goal | 2.47e-20 | 1.42e-02 |
| Uncertainty | 2.94e-03 | 7.33e-01 |
| Goal x uncertainty | 1.85e-07 | 3.84e-10 |
| Depression score level | 1.13e-05 | 5.30e-05 |

**Fig 4. Computational model of arbitration explains subjects' behavioral patterns associated with depression symptom.** (a) Proportion of actual choice of the agent [26,30], based on human subjects' data (left) and averaged data of 100 simulations of the computational model fitted on each subject (middle). The healthy and subclinical depression group were defined by the standard cutoff criteria of the CES-D score 16 [31,32]. It is the proportion of the right action (R) as a function of choice probability, computed based on the value difference between the two choices (R and L). The key effects found in the subjects' data were replicated in the simulation data. (b) Choice consistency, a behavioral measure associated with model-free learning. It is defined as the proportion of making the same choice as in previous trials. The significant correlation between choice consistency and the depression symptom (CES-D) in subjects' data (left scatter plot) is replicated in the model simulation data (right scatter plot). The results were replicated in terms of the effect of the key task variables (goal and uncertainty) on choice consistency (error bar plots and the statistical test results; both the main and interaction effects were replicated). (c) Choice optimality, a measure of the extent to which a subject's choice reflects an optimal policy. The significant correlation between choice optimality and the depression symptom (CES-D) in subjects' data (left scatter plot) is replicated in the model simulation data (right scatter plot). The results were also replicated in terms of the effect of the key task variables on choice optimality (except for the uncertainty effect).

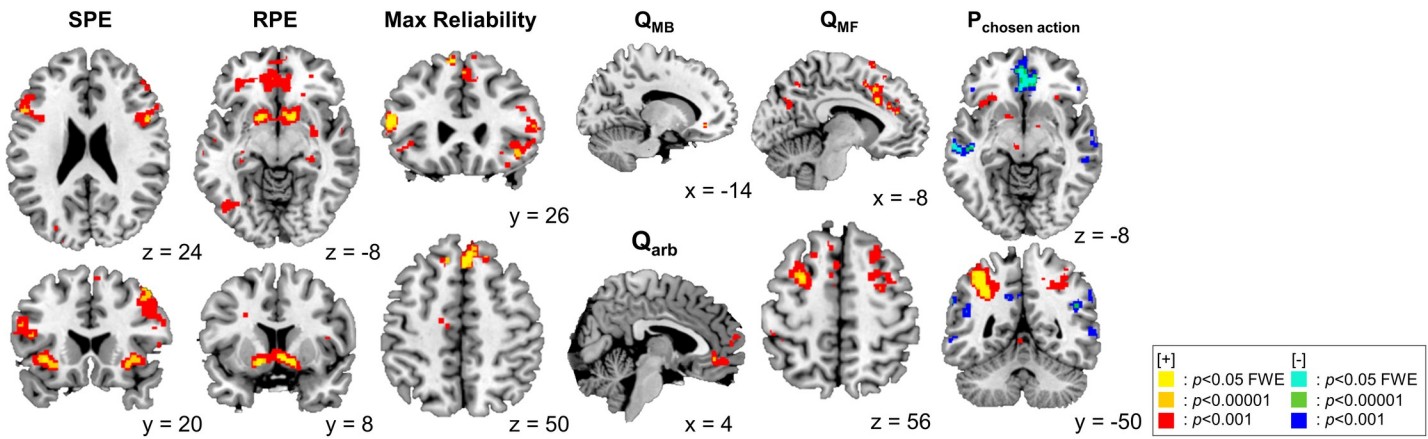

**Fig 5. Neural correlates of dynamic arbitration control.** Prediction error, value, reliability, and action choice probability signals from the proposed model are shown as colored blobs. SPE and RPE refer to state prediction error and reward prediction error, respectively. Max reliability refers to the reliability of whichever system had the highest reliability index on each trial (= max(Rel_MB, Rel_MF)). $Q_{MB}$ and $Q_{MF}$ indicate the chosen value from the model-based and model-free system. $Q_{arb}$ refers to the difference between integrated chosen action value and unchosen action value. $P_{chosen\ action}$ refers to the probability assigned to the chosen action. See more detailed information in Table A, B, and C in S1 File.

globus pallidus, FPC, supplementary motor area, and thalamus (all p<0.05 FWE corrected) are positively correlated with the probability value of the chosen action, referred to as the output value of the softmax function [41]. This finding is consistent with previous findings indicating stochastic action selection [42]. Other brain areas, such as the orbitofrontal cortex, superior temporal gyrus, middle temporal gyrus, supramarginal gyrus (p<0.05 FWE corrected), medial PFC, and superior frontal gyrus (p<0.05 FWE corrected) are negatively correlated with the chosen action probability. The negative encoding of stochastic action selection in the medial cortex replicates previous findings, as the region has been implicated in the valuation of counterfactual choices [43] (Fig 5 and Table C in S1 File).

## Neural effects of subclinical depression on model-based and model-free learning

To fully explore how depression affects the neural computations underlying model-based and model-free RL, we examined the relationship between the individual depression score and neural representations in each brain region implicated in model-based and model-free RL. We found evidence indicating the effect of subclinical depression on RL in multiple brain areas encoding prediction errors. The correlation coefficient between left insula activation and the SPE ([−36,20,−4], z = 4.49), which represents the efficiency of neural encodings of the SPE in the left insula, was inversely proportional to the depression score (estimated correlation coefficient = -0.396, $p$ = 0.037; Fig 6A). We also found a significant negative correlation between the neural efficiency for encoding the RPE in the bilateral caudate (left: [−4,6,−4], z = 4.50, right: [4,8,−4], z = 5.30) and the individual depression score (estimated correlation coefficient = -0.412/-0.376, $p$ = 0.029/0.049 for the left and right, respectively; Fig 6B). These findings directly demonstrate how subclinical depression affects value updates for goal-directed and habitual learning.

## Neural effects of subclinical depression on arbitration control

Next, we tested for the neural effects of subclinical depression on arbitration control. The first evidence we found is that the individual depression score is significantly correlated with the

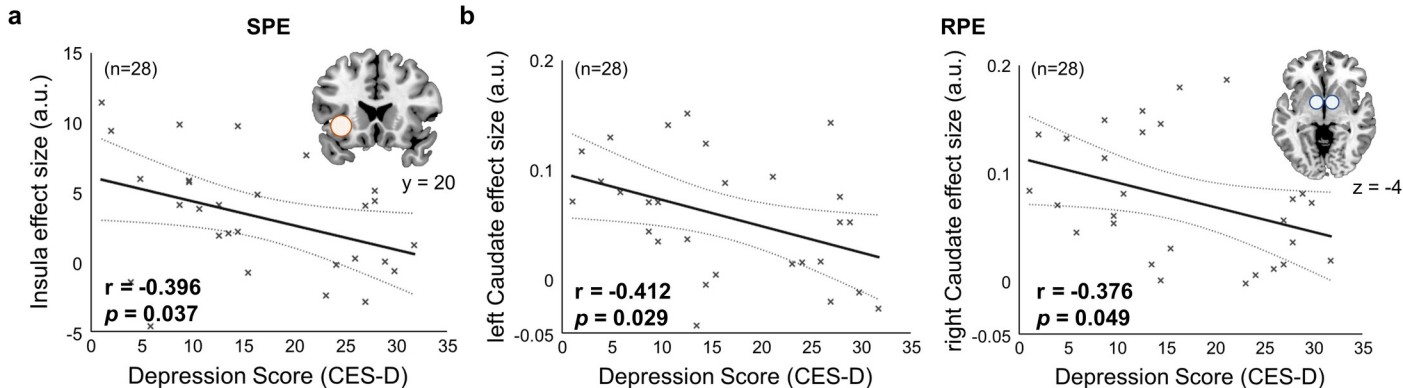

**Fig 6. Parametric effect of subclinical depression on model-based and model-free learning.** (a) Effects of subclinical depression on neural encoding of state prediction error (SPE) information (n = 28). The shaded circles represent seed regions for which parameter estimates of the GLM analysis were extracted. The seed region is the left insula, and the parameter estimates were extracted from our GLM analysis which regressed the SPE signal against the BOLD response ([−36,20,−4], z-score = 4.49; Fig 5). The estimated effect size for the left insula is negatively correlated with the depression score. (b) Effects of subclinical depression on reward prediction error (RPE) response (n = 28). The estimated effect size of RPE for bilateral caudate (left: [−4,6,−4], z-score = 4.50, right: [4,8,−4], z-score = 5.30) is negatively correlated with the depression score. a.u. stands for arbitrary units.

learning rate for model-free reliability estimation, the key variable required to quantify the reliability of predictions made by the model-free learning strategy based on the RPE (Spearman's rho = 0.335, $p$ = 0.007; Fig 7A).

This finding offers a theoretical prediction that depressive symptom entails over-sensitivity to the RPE, making arbitration control more sensitive to the prediction of the model-free system (Fig 7B). Critically, this theory can be confirmed by testing the following two hypotheses: (1) the brain regions previously implicated in mediating arbitration control focus on information about the reliability of the model-free RL in the subclinical depression group, compared to the healthy group, leading to the disruption of encoding the reliability information of the RL system controlling behavior at the moment (i.e. max reliability), leading to (2) the disruption of neural computation underlying arbitration control.

To formally test the first hypothesis, we ran an MVPA for the bilateral ilPFC and FPC. This analysis quantifies the amount of information concerning the reliability of the model-free system embedded in these brain areas. We used a support vector machine, an optimal shallow neural network that can quantify the amount of information with minimal risks of overestimation, to conduct a binary classification of model-free reliability (high vs. low; upper/lower 33rd percentile threshold) and compared the prediction performance of the control and subclinical depression groups (the same groups used in Fig 4).

We found that the prediction performance of model-free reliability in the bilateral ilPFC and FPC was significantly higher in the subclinical depression group than in the control group (Fig 7C; one-way ANOVA; $F_{1,26}$ = 4.57 [$p$ = 0.042], $F_{1,26}$ = 4.33 [$p$ = 0.047], $F_{1,26}$ = 4.89 [$p$ = 0.036] for the left and right ilPFC and FPC, respectively). On the other hand, the same analysis found no significant inter-group differences in model-based reliability signal or max reliability signal (Table D in S1 File).

This finding is linked to the second hypothesis that subclinical depression disrupts neural processing pertaining to arbitration control between model-free and model-based RL, we conducted a GLM analysis with the max reliability signal. We found that the effect size (parameter estimates from the GLM analysis) of the max reliability signal for FPC ([8, 44, 40], z = 3.83) was negatively correlated with the depression score (estimated correlation coefficient = -0.441, $p$ = 0.019; Fig 7D, right). Moreover, the effect sizes of the max reliability signal for the bilateral ilPFC ([−52,26,16], z-score = 4.48; [42,20,−8], z-score = 3.61) and FPC ([8, 44, 40], z-score =

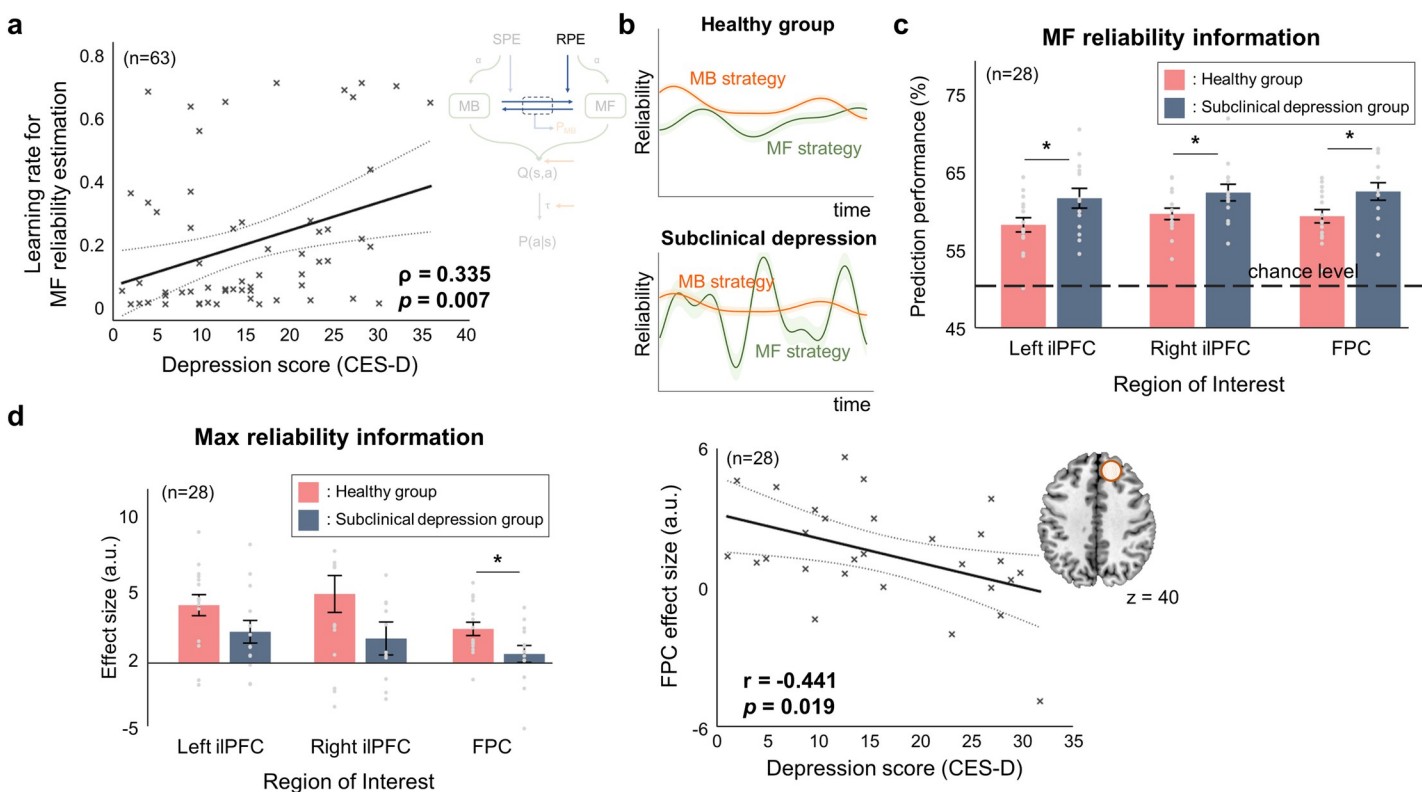

**Fig 7. Depression impacts on prefrontal arbitration control.** (a) Relationship between the depression score and the learning rate parameter for reliability estimation of the model-free system (n = 63; the total number of subjects, including 35 who participated behavioral experiment only and 28 who were also scanned with the fMRI). Individuals with a higher depression score tend to exhibit a higher learning rate, indicating that their reliability estimation for the model-free system is very sensitive to reward prediction error (RPE). (b) Illustrative examples of reliability changes of people with a low ("healthy group") and a high depression score ("subclinical depression group"), each of which is associated with a low and a high learning rate for reliability estimation, respectively. The subclinical depression group shows rapid changes in MF reliability due to higher learning rate. (c) Quantification of the amount of reliability information of the model-free system for the healthy (n = 15) and depressive (n = 13) group (MVPA analysis). The MVPA analysis with these three seed regions reveals that the amount of information about the model-free reliability was significantly higher in the subclinical depression group in all three regions (one-way ANOVA). Asterisk (*) indicates significant difference at the 0.05 level. (d) (Left) The parameter estimates of the Max reliability for the bilateral inferior lateral prefrontal cortex (ilPFC) ([−52,26,16], z-score = 4.48; [42,20,−8], z-score = 3.61) and FPC ([8,44,40], z-score = 3.83). Their effect sizes tend to be smaller in the subclinical depression group. (Right) Relationship between the depression score and the estimated effect size of Max reliability for frontopolar cortex (FPC), the brain area implicated in arbitration control. The estimated effect size of the Max reliability signal for FPC is negatively correlated with the depression score.

3.83), the brain areas previously implicated in arbitration control [26,35,44], tended to be lower in the subclinical depression group (CES-D score≥16) than in the healthy group (CES-D score<16) (Fig 7D, left; one-way ANOVA; $F_{1,26} = 2.76$ [$p = 0.108$], $F_{1,26} = 2.93$ [$p = 0.099$], $F_{1,26} = 5.18$ [$p = 0.031$] for the left and right ilPFC and FPC, respectively).

Taken together, these three evidences strongly support our arbitration control hypothesis that depressive symptom is associated with increased sensitivity to RPE, leading to unstable arbitration in which the reliability of predictions of the model-free system becomes predominant and the reliability of predictions of the model-based system becomes less influential.

## Neural effects of subclinical depression on value–action conversion

Finally, our computational model allows us to account for how the subclinical depression influences the ability to convert the value, which is learned by arbitration control, into actual choice. In our model, this ability is parameterized as an exploitation sensitivity. For example, exploitative and explorative choices are associated with high and low exploitation sensitivity, respectively. We found that the exploitation parameter for model-based RL is negatively

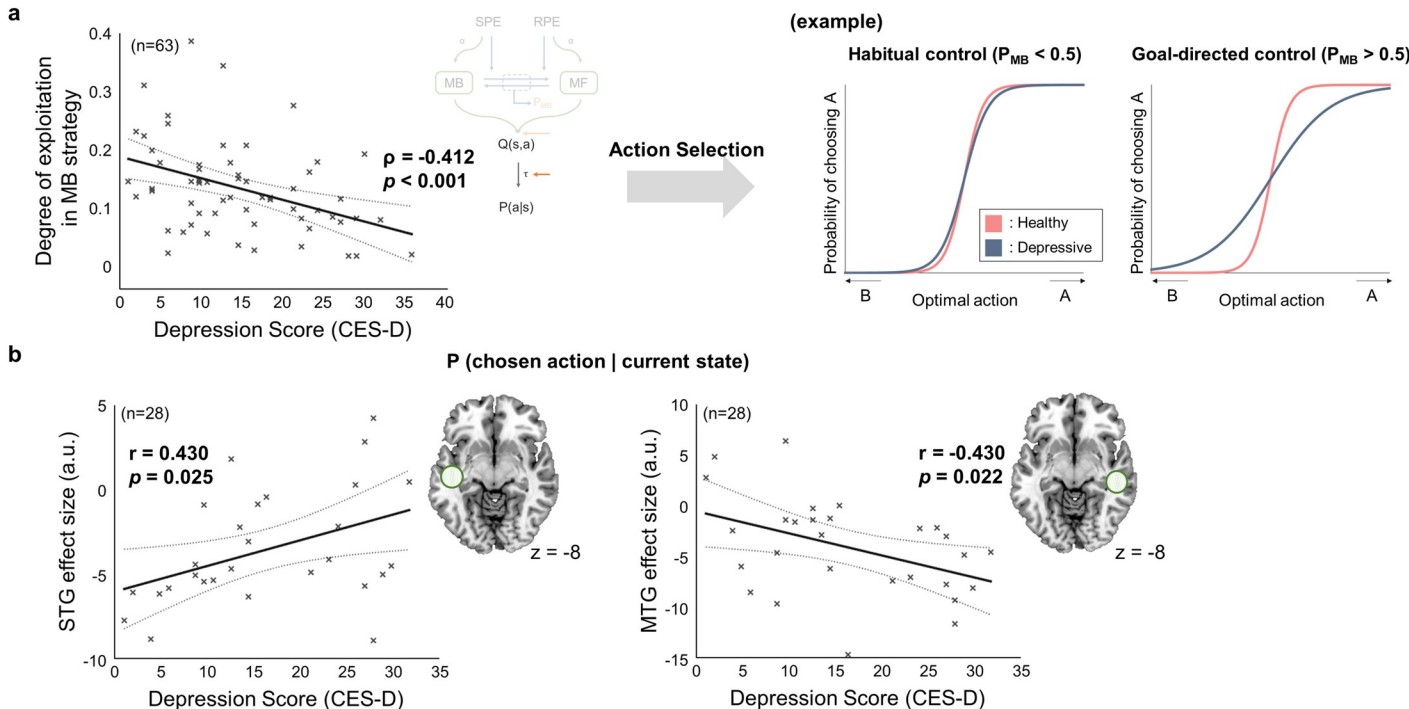

**Fig 8. Parametric effect of subclinical depression on value-action conversion.** (a) Effects of subclinical depression on model parameters (n = 63; the total number of subjects, including 35 who participated behavioral experiment only and 28 who were also scanned with the fMRI). (Left) Relationship between the depression score and the degree of exploitation in model-based (MB) strategy ($\tau_{MB}$). The MB exploitation parameter decreases as the depression score increases. (Right) Examples illustrating exploitation parameter effects on action selection. Shown are the softmax functions that convert an action value into a choice probability value, between the healthy and the subclinical depression group for goal-directed and habitual control. Compared to the healthy group (pink), the subclinical depression group makes more exploratory choices especially as they rely more on the model-based system (blue). (b) Relationship between the depression score and the parameter estimate of the probability of selecting chosen action for the two seed regions, left Superior Temporal Gyrus (STG) ([–46,–20,–8]; z-score = 4.21) and the right Middle Temporal Gyrus (MTG) ([56,–32,–8]; z-score = 3.55). The effect size of left STG and right MTG increases and decreases with the individual severity of depressive symptom, respectively.

correlated with the individual depression score (Spearman's rho = -0.412, $p < 0.001$; Fig 8A, left), indicating that subjects with higher depression scores exhibit more exploratory choices when their choices are guided by model-based RL (Fig 8A, right). Note that by factoring the individual bias of model-based control into the arbitration model as done in the original arbitration model (the free parameter of the transition rate function [26,29], we precluded the possibility that this exploratory behavior is ascribed to the individual prowess of using model-based strategies.

In the subsequent neural analysis, we explored the relationships between the depression score and the neural representations of value–action conversion. The parameter estimates of the probability value of taking the chosen action are significantly correlated with the depression score in two brain regions. The parameter estimates in the two seed regions—the left superior temporal gyrus (STG; [–46,–20,–8], z = 4.21) and right middle temporal gyrus (MTG; [56,–32,–8], z = 3.55)—are positively and negatively correlated with the CES-D score, respectively (correlation coefficient = 0.430, $p = 0.025$ for STG; correlation coefficient = -0.430, $p = 0.022$ for MTG; Fig 8B).

## Discussion

By combining a computational model allowing for sub-optimality in learning and decision-making, a model-based fMRI analysis, and a MVPA, the present study fully characterizes how

subclinical depression influences the different levels and stages of RL: value learning, prefrontal arbitration control for value integration, and value–action conversion. On the other hand, the expected state-action value itself is seemingly not influenced by the subclinical depression, which is consistent with the previous finding that the expected value is intact in MDD [45]. We found that subclinical depression has a parametric effect on neural representations of prediction error for model-based and model-free systems, respectively, explaining how subclinical depression hampers value computation ability. Another intriguing finding is that the brain areas implicated in arbitration control, bilateral ilPFC and FPC, become more sensitive to the predictions of the model-free system in people with higher depressive symptom, indicating that depressive symptom disrupts the balance between goal-directed and habitual control. We also found that subclinical depression increases the tendency to make exploratory choices during model-based control, but not during model-free control. The results of the impaired SPE representation and the increased amount of MF information in the arbitration center in higher depressive symptom are novel findings.

## Computational theory of suboptimal model-based and model-free control

The present study's computational model of dynamic arbitration control of model-based and model-free RL allows us to explore the full parametric effects of subclinical depression on prefrontal goal-directed and habitual control. Although mounting evidence suggests that prediction uncertainty might be a key variable for prefrontal arbitration control [6,26,33], little is known about the computational reasons why people with depression tend to exhibit behavioral biases towards either goal-directed or habitual behavior. Addressing this issue involves a few challenges. First, the fact that both the model-based and model-free learning guide value-based decision making it necessary to incorporate the interaction between the two types of learning into the computational model. Second, there is no guarantee that a rational arbitration control model is flexible enough to explain the individual variability associated with depression. Third, exploring a depression-specific model based on the assumption that depression follows a computational regime that substantially deviates from rational decision-making may enable us to explain severe depression, but cannot explain a continuum extending from a healthy to a severely depressed state.

To fully address these issues, we considered a computational model of dynamic arbitration control allowing for individual variability in suboptimal learning and decision making. Intriguingly, we found that the influence of prediction uncertainty is not confined to value integration [26], but extends as far as value–action conversion. This also allowed us to test the full effect of subclinical depression on decision-making at different computation levels: model-based and model-free reinforcement learning, arbitration control for value integration and value–action conversion. To validate whether the chosen model correctly learns the human behavior, we conducted two types of post-hoc analyses: behavior recovery and model parameter recovery. The results of the behavior recovery analysis showed that the free parameters of our model successfully encapsulates the essence of subjects' choice behavior (Figs 3C and 4). The model parameter recovery analysis revealed that the model parameters are well-estimated without overfitting, especially for the two key parameters that are found to encode the effect depressive symptoms on behavior (Fig B in S1 File). This fully establishes a link between the model and behavior. Future works can use this model to predict the severity of subclinical depression as this model is fitted with individual behavioral data.

An open question concerns that in simple reward learning tasks, such as Daw's [6], there may not be enough room for agents to actually use distinctly different two learning strategies [46]. This raises alternative possibility that behavior driven by suboptimal model-based

learning can be confused with model-free learning. That being said, our task design was optimized for encouraging use of both strategies, as indicated by the early study demonstrating engagement of neural systems associated with both model-based and model-free [26,30,47]. For example, in the specific goal condition, choice consistency of model-free learning is distinctly different from that of suboptimal model-based learning. Furthermore, the high uncertainty condition of our experiment is intended to plummet the performance level of the suboptimal model-based learning, lower than that of the model-free counterpart. Further study should explore behavioral indicators that can gauge optimality of model-based learning in a way that can distinguish between model-free learning and suboptimal model-based learning.

## Effects of subclinical depression on reinforcement learning

Our study found that subclinical depression has a parametric effect on the neural representations of the two distinct types of prediction errors associated with model-based and model-free RL: SPE and RPE. The neural analysis revealed that depression scores were correlated with an attenuation of the SPE signal in the left insula and the RPE in the bilateral caudate.

Dopamine is crucial for both model-free and model-based RL. Numerous previous studies have reported dopamine's role in guiding the RPE [48,49], and a recent finding discussed the essential role of dopamine in stimulus-stimulus associative learning [50], implicating the involvement of dopamine in SPE representation. Depression is characterized by decreased dopamine levels [51,52], which may impair learning in both model-free and model-based systems. In fact, RPE signals in depression have reportedly been reduced in various experimental conditions (both Pavlovian learning [13] and instrumental learning [14–17]). Our study not only corroborates previous findings concerning RPE deficits in depression, but also further suggests that depression may impact neural representations of SPE.

Another open question that merits further investigation is potential influence of depression on different types of cognitive processes. For example, there is commonalities and differences between our results with a learning task and Rutledge's work [45] focused on a non-learning task structure. In Rutledge's work, intact RPE representation is shown in MDD. We believe that the exploring the difference between the learning and non-learning case in terms of depression would be an interesting future direction.

The effect of subclinical depression on neural encoding of reward prediction error information was significant bilateral caudate, but the effect on state prediction error encoding was found to be significant in the left insula, albeit being tested bilaterally (Fig 6). This might be related to a few open questions, such as depression-specific lateralization of brain functions or biological characteristics of mild and situational depression that may not be clinically diagnosed [53–58]. We should note that given the fact that the severity of depression in our subjects is modest, it is very challenging to find the behavioral and neural effect. We think it is worthwhile to report these results of Fig 6, along with other neural effects (Figs 7 and 8) in the main text, as they merit further investigation.

Note that our behavioral results are fully consistent with the results of our model-based parametric GLM analyses (Fig 6). First, we found that the extent to which the insula encodes the state-prediction error signal, the key variable for model-based learning, is negatively correlated with the depression score (Fig 6A). This result corroborates the above behavioral finding (Fig 2B) showing that subclinical depression might have a negative impact on model-based learning. Second, we found that the extent to which the caudate encodes the reward-prediction error signal, the key variable for model-free learning, is negatively correlated with the depression score (Fig 6B). This result corroborates the above behavioral finding (Fig 2C) showing that subclinical depression might have a negative impact on model-free learning.

## Effects of subclinical depression on arbitration control of reinforcement learning

One interesting prediction of the model is that individual depression score is positively correlated with the parameter value for controlling the learning rate for updating model-free reliability based on the RPE, indicating that reliability estimation for the model-free strategy is very sensitive to RPE changes. This suggests that, in people with high CES-D scores, arbitration control might be predominantly driven by the model-free reliability signal, rather than by a fair comparison of the model-free and model-based reliability signals. This result is consistent with the previous finding that depressive symptoms are relevant to the elevated reflexive behaviors [12]. We explored this possibility through a combination of a general linear model (GLM) and MVPA.

Our GLM analysis showed the negative effect of the subclinical depression on the neural representations of arbitration control in the prefrontal cortex. In bilateral inferior lateral PFC and frontopolar cortex, the brain areas reportedly encoding the key variable for arbitration control [26], neural representations tend to be weaker in the high CES-D score group. However, the effect was only significant in the FPC, not in the ilPFC. Critically, the subsequent MVPA shows that the amount of reliability information of the model-free system is significantly higher in the high CES-D score group. These findings theoretically implicate that subclinical depression may hinder the PFC's ability to estimate the reliability of each learning strategy from the corresponding prediction error.

It appears that the decrease in the overall performance as a function of the depression score in our model-free analysis (Fig 2A) can be explained by the model-based MVPA analysis (Fig 7) meant to examine the performance of prefrontal arbitration control. Specifically, we found that the depression score is negatively correlated with the level of stability of the reliability estimation process (Fig 7A). In other words, the subclinical depression group showed less stable reliability estimation (Fig 7B), undermining stability of the competition between model-based and model-free learning strategy (Fig 7C and 7D). This means that subjects with a lower depression score are more likely to use a suboptimal strategy that is less suitable for the current context (e.g., goal or state-transition uncertainty), leading to the impairment of overall learning performance (Fig 2).

One potential advantage of our arbitration control model-based approach is that it allows us to examine how the brain deals with speed-accuracy tradeoff [59–61]. In principle, optimal arbitration control should tradeoff the accuracy when the agent was tempted to quickly perform the task, while compromising the speed when it is sufficiently motivated to maximize the task performance [62]. Although it is an interesting issue worth examining, the experimental paradigm we use in this study was not intended to investigate this tradeoff, so it should be left for future research.

## Effects of subclinical depression on value–action conversion

The present study also provides a computational and neural account of how subclinical depression causes sub-optimal action selection. Our computational model predicts that subclinical depression increases the tendency to make exploratory choices during model-based control, rather than model-free control.

Value-action conversion is an another potentially interesting factor that may influence the overall performance (Fig 2A). The negative relation between the degree of exploitation of the model-based learning strategy and the depression score (Fig 8A) suggests that the value information learned by the model-based strategy is less likely to be reflected in actual choices, essentially leading to the decrease in not only overall performance (Fig 2A and 2B) but also consistency of choices (Fig 2C).

This finding could also clarify the two conflicting views of choice consistency behaviors in depression [63,64]. Beever et al. (2013) found no significant difference in exploration pattern in a reward-maximizing task between a healthy and a depressed group. Blanco et al. (2013), on the other hand, found that a depression group tended to explore more. This conflict might be attributable to differences in task structure. Beever's study used a task with a relatively stable environmental structure, such that people performed tasks relying on the model-free system. This is consistent with our view that depression has a relatively weak influence on exploration during model-free control. However, the reward structure used in Blanco's study encouraged more frequent policy changes, accommodating the need for model-based control. This is also consistent with our view that exploratory choice behavior becomes more pronounced during model-based control.

The neural results of the present study, which show that the STG response is higher in people with higher depressive symptom, are fully consistent with previous finding that STG response increases when people switch to other options rather staying [65]. Our results address not only the implication for the role of STG in exploration, but also how subclinical depression influences exploration at the neural level.

Our finding that the degree of exploitation decreases as CES-D score increases (shown in Fig 8) explains why reward sensitivity is reduced in people with depression. A decreasing degree of exploitation decreases the tendency to convert a learned policy into an actual choice, reducing the efficiency of translating changes in the reward structure into changes in actual choices. This is also consistent with the view that our model's degree of exploitation parameter can be interpreted as reward sensitivity [15,66]. In addition to supporting existing evidence of declined reward sensitivity in depression [27,67,68], the present study advances the view by proposing that this tendency becomes stronger during model-based control.

It would be interesting to further establish the relationship between depression score and valuation, including value (Q-value) and action-value conversion ($P_{chosen}$). Though it is seemingly a simple question, the underlying issues are more complicated than it sounds. First, each subject experiences different episodes of events caused by different choices and uncertainty of the environment. This makes it hard to directly relate subject experience to the depression score (e.g., averaged Q-value of average number of choices). Second, these are a few potential confounding factors contributing to choices, such as individual performance, choice bias, etc. which may or may not be associated with depression. Our study avoids the above issues by focusing on model-based/parametric analyses, which aim to investigate the depression effect on the learning process underlying choices. Specifically, our computational model of arbitration control, in which its parameters were individually fitted to subject's data, enables us to separate out the characteristics of choice behavior from those associated with various individual variability, ranging from learning performance and choice bias to the strategy bias towards model-free/model-based learning. Model fitting by itself can be viewed as the process of factoring out individual biases from the data.

## Potential clinical applications and limitations

The present findings suggest how subclinical depression influences goal-directed and habitual control in the prefrontal–striatal circuitry. The full characterization of the effects of subclinical depression on different stages of learning and decision-making creates possibilities for various clinical applications. First, our neural results would give a clue why brain stimulation techniques such as repetitive transcranial magnetic stimulation to the frontopolar cortex [69] or deep brain stimulation to the ventral striatum [70] are effective in alleviating depressive symptoms. Our results showing reduced RL signal representations in these regions of a subclinical

depression group raises the expectation that the electrical stimulation may help restoring neural functions associated with RL. In addition, regaining reward learning could also alleviate the anhedonia symptoms [4]. Second, our finding suggesting that over sensitivity to the MF system in subclinical depression group disrupts the balance between the MF and the MB system may explain why the MB system is implicated in emotional regulation [11].

The recent arbitration theory predicts that successful arbitration between the two systems makes specific reliability signals diminish [26]. Our results suggest that subclinical depression disrupts this process; in depressive individuals, the brain regions engaged in the arbitration process still have information about the MF system (Fig 7A and 7B), and this may be attributable to the elevated PE sensitivity in MF system (Fig 7). This theoretical idea creates a possibility that behavioral therapy to reduce sensitivity to reward prediction errors might help people with subclinical depression stably estimate the MF reliability over a long period of time, thereby regaining a balance between the two systems. As the dynamic interaction between MB and MF in our study provides a means to better understand depressive symptoms, further study may consider applications that incorporate MB-MF tradeoffs into the diagnosis of depression. Recent studies have successfully modeled the tradeoffs in this context, such as cost-benefit or speed-accuracy, where MB value computation is more costly but also more accurate (and ultimately provides more rewards) than MF computation [59,71].

Intriguingly, we found the parametric effect of depression on learning and decision-making in a relatively young age group (average = 22.8 yrs). Considering the onset of MDD is approximately 25 to 45 yrs [72], our study offers possibilities for not only investigating how subclinical depression transitions to MDD, but also developing clinical applications for the early diagnosis of MDD.

There are several limitations in this work. First is a lack of measurements of other pathological symptoms related to depression, such as anxiety or personality disorder. In addition, we did not assess whether depressed participants in our research based on the CES-D criteria actually suffered from clinical levels of depression or diagnosed by the psychiatrist. We focused on the individuals with subclinical depression, had a relatively small sample size of elevated depressive symptoms, and lacked of analyses investigating the potential moderating effects of symptom profiles. Further investigation should extend our findings to elevated depressive symptoms, eventually Major Depressive Disorder.

Our work combines computational model-based and machine learning approach to investigate the cognitive dysfunction in subclinical depression. We believe that our findings not only help us establish a rich connection between impaired decision making and subclinical traits of depression, but also raise the possibility for investigating cognitive impairments extending from a normal to subclinical psychiatric diseases.

## Methods

### Ethics statement

Every participant provided written consent to the experimental protocols which were approved by the Institutional Review Board (IRB) of the Korea Advanced Institute of Science and Technology (KAIST).

### Participants

65 right-handed Koreans (28 females; mean age of 22.8±3.8) participated in the study. Participants were recruited from the local society through the online announcement. Only 28 subjects (13 females) were scanned with fMRI during the task. Both the behavior-only subjects and the fMRI subjects performed the tasks under the identical experimental setting except the

fMRI group conducted the experiment inside the scanner. Two subjects whose total accumulated rewards were below the chance-level (mean amount of rewards with 10,000 random simulations) were excluded from the analysis. Thus, a total of 63 behavioral data and 28 fMRI data were left for the analysis. No subjects had a history of neurological or psychiatric diseases.

### Rationale for using the CES-D score

To assess the depressive level of individuals, people were instructed to complete the Center for Epidemiologic Studies Depression (CES-D) questionnaires [73] before the experiment. We collected both CES-D and PHQ-9 data initially, and chose to use the CES-D score on the following basis.

The first rationale for the use of CES-D is that this score has been known to be reliable [73,74]. Our initial analyses suggested that statistical results from the PHQ-9 did not add up well (inconsistency in results from different model-based analyses), making us undermine the reliability of this measure. One reason for insufficient reliability of the PHQ-9 might be ascribed to the fact that it is designed to be easily administered and scored; the length of PHQ-9 questionnaires is half that of CES-D.

Second, it is known that the CES-D is effective in capturing short-term symptoms [75]. This characteristic makes it easy for us to assess the effect of subclinical depression, for which development of symptoms may not be highly persistent.

Third, to maintain coherence with other related works, we conform to the convention of examining the effect of depression on decision making. For example, two studies [63,76] used the CES-D score to examine the effect of depression symptoms on decision making. Note that they used a learning task with environmental uncertainty, which falls into the same category as our task.

### Experimental design

We used the sequential two-stage Markov decision task proposed to dissociate model-based and model-free learning strategies [26]. In this task, subjects make a binary choice (either left or right) and proceed to the next state with a certain probability. When the next state appears on the screen, participants make another choice. The two consecutive choices are followed by a transition to an outcome state. Each state is represented by a different fractal image. The first image (state) is always the same, indicating the beginning of each trial. Subjects perform 100 trials in the pre-training session to learn the structure of the task. Four main sessions with an average of 80 trials per session follow the pre-training session. In the pre-training session, participants are instructed to learn the state-space, including outcome states associated with the coins. In the main sessions, they were instructed to collect as many coins as possible; they were notified that they will earn additional money proportional to the accumulated rewards they acquire during the main sessions. In each trial, state transition occurs between 1 to 4 seconds after making a choice; transition timing was sampled from a uniform distribution [1,4]. The intertrial interval was also drawn from the same distribution. The reward picture appeared for 2 seconds. The pre-training session was intended to provide subjects an opportunity to acquire a reasonable amount of knowledge about the state space. It took 12 to 20 minutes. The main session took 12 to 15 minutes per session. Participants took approximately 80 minutes to perform the task. Note that both experience in the pre-training and main session were incorporated into the model fitting to fully accommodate individual variability.

The task consists of two conditions: a specific-goal condition and a flexible-goal condition. The goal condition is indicated by the color of a box at the beginning of each trial. In the specific-goal condition, participants are presented with a box with a specific color (red, blue, yellow). A monetary reward is given only when the coin color matches with the color of the given

box. In the flexible-goal condition, on the other hand, participants are given a white coin box with which all types of coins become redeemable. The specific-goal condition and flexible-goal condition are designed to promote model-based and model-free learning, respectively. At the end of each trial, subjects are told the amount of money they earned.

We used two types of state-transition probability to control the level of uncertainty of the environment. This manipulation was intended to dissociate the model-based and model-free control and also to preclude excessive use of a specific strategy in some of the goal conditions. The state-transition probability (0.5, 0.5) and (0.9, 0.1) is intended to implement the highly-uncertain and relatively less uncertain environment, respectively, each of which one learning strategy is preferred over the other. The state-transition probability value was applied to all transitions within each trial (the first to the second state and the second to the outcome state). Participants were not informed about the state-transition probabilities in the task, but they were told that these might change during the course of the task. The color of the coin box appears on the screen to indicate the goal condition (flexible-white/specific-red or blue or yellow). The four types of block (2 goal-conditions x 2 uncertainty conditions) are presented in pseudo-random order. Transitions between 4 experimental conditions allow participants to flexibly allocate behavioral control over behavior to model-based and model-free learning.

The low-uncertain block lasts 3–5 trials, whereas the high-uncertain block consists of 5–7 trials, enabling participants to learn the underlying state-transition probability (Fig 1). Note that the length of each type is not meant for participants to fully learn the transition-probability, but for eliciting the changes of the reliability of the predictions of the model-based and the model-free strategy.

## Model-free behavioral measures

Our model-free behavioral analysis aims to explore pure behavioral effects of subclinical depression on choice behavior. In doing so, we used distinctly different three behavioral measures, each of which is associated with overall learning performance, performance of model-based learning, and performance of model-free learning. Specifically, the first behavioral measure, accumulated reward (Fig 2A), is a conventional measure to assess behavioral performance. This helps us evaluate the relationship between overall learning performance and the CES-D score. The second behavioral measure, the choice optimality (Fig 2B), is a behavioral read-out for model-based control [26,30], whereas the third measure, choice consistency (Fig 2C), is a behavioral read-out for model-free control [26]. Definitions and justifications of each measure are provided in the following two sections.

The choice optimality measure quantifies the extent to which participants on a given trial took objectively the best choice had they complete access to the task state-space, and a perfect ability to plan actions in that state-space. It is based on the choice of the ideal agent assumed to have a full, immediate access to information of the environmental structure, including state-transition probability and goal changes. It is defined as the degree of match between subjects' actual choices and an ideal agent's choice corrected for the number of available options. To compute the degree of choice match between the subject and the ideal agent, for each condition, we calculated an average of normalized values (i.e., likelihood) of the ideal agent for the choice that a subject actually made on each trial. The choice optimality value would have a maximum/minimum value if a subject made the same/opposite choice as the ideal agent's in all trials, regardless of complexity condition changes.

Owing to the fact that the ideal agent's behavioral policy is not affected by the variability of such experimental variables, it serves as a good proxy for assessing the degree of participants' engagement in model-based control. In principle, provided that the model-based agent has

complete knowledge of the state-space and no cognitive constraints, it will always choose more optimally than a model-free agent. A separate computer simulation in one of our recent studies [30] has demonstrated that the choice optimality of the model-based agent is significantly higher than that of model-free agent.

Choice consistency, also known as choice repetition, is a conventional behavioral measure used to quantify insensitivity to changes in the environmental structure (one of the key characteristics of model-free RL), works well for conventional two-step task paradigms in which the environment is stable for a certain period [47,77]. For example, when one outcome is associated with a relatively large amount of reward, a model-free agent is likely to repeat the same choice regardless of changes in environmental structures. It is known to account for habitual behavior. These characteristics make it easy to evaluate behavioral bias attributable to model-free control.

However, we are aware of its limitation in that it may not be suitable for highly dynamic task design which necessitates a mixed contribution of model-based and model-free control, so we used model-based analyses (Fig 3) to fully investigate separate contributions of model-free and model-based control on behavioral performance and its relationship with depression score (Figs 6, 7, and 8).

## Computational models

The computational model of this study is motivated by the previous arbitration control hypothesis that prediction uncertainty of the model-based and model-free RL is a key variable to guide value integration of the two corresponding systems [26]. The model consists of three processes: value learning, arbitration, and action selection.

In the value learning stage, both a model-based and model-free system learn action values for each state. A model-based system uses state prediction error (SPE = 1-expected transition probability) to update the state-action-state transition probability ($\Delta T(s,a,s') = \kappa(1-T(s,a,s'))$, $\kappa$: learning rate), by using a FORWARD learning [33] and learns action values by combining the learned state-action-state transition probability and reward in the outcome state.

$$Q_{MB}(s, a) = \sum_{s'} T(s, a, s')\{r(s') + \max_{a'} Q_{MB}(s', a')\}$$

(T(s,a,s'): the probability of transition to state s' when taking action a from state s, r(s'): reward in state s', Q(s,a): state-action value)

For a model-free system, on the other hand, the state-action value learning is based on RPE (RPE = actual value-expected value). It is implemented with a SARSA algorithm [5].

$$Q_{MF}(s, a) = Q_{MF}(s, a) + \alpha(r(s') + Q_{MF}(s', a') - Q_{MF}(s, a))$$

($\alpha$: learning rate)

It is assumed that the brain tracks the reliability of the two systems to combine their value estimates. The reliability estimate is based on the two types of prediction errors, SPE and RPE, to implement the fact that either system with high prediction error is unreliable. Specifically, the reliability of the model-based system was estimated as an inverse of the index of dispersion with a hierarchical empirical Bayes method using the history of the SPE. Note that this specific implementation with a hierarchical Bayes model is based on earlier studies on the probabilistic computation of the neural population [78,79].

$$\chi_{MB} = \frac{\chi_0}{\chi_0 + \chi_1}, \chi_i = \frac{E[\theta_i|D]}{Var[\theta_i|D]}$$

($\theta_0$: probability of zero SPE, $\theta_1$: probability of non-zero SPE, D: history of SPE)

On the other hand, the reliability of the model-free system was implemented with the Pearce-hall associability rule using an unsigned RPE rather than the complex Bayesian estimation.

$$\chi_{MF} = \frac{RPE_{max} - \Omega}{RPE_{max}}$$

($\Omega$: absolute RPE estimator, $\Delta\Omega = \eta(|RPE|-\Omega)$, $\eta$: learning rate, $RPE_{max}$: the upper bound of RPE which is used to normalize the reliability)

These estimated reliability signals were then used to guide the competition between the two systems, which is implemented with a dynamic two-state transition model. The output of this model is a model choice probability ($P_{MB}$), used as the control weight for value integration of the two systems.

$$\frac{dP_{MB}}{dt} = \beta(1 - P_{MB}) - \gamma P_{MB}$$

($\beta$: transition rate of MF→MB; it is the function of the prediction reliability of the MF system based on the averaged RPE ($\chi_{MF}$). $\gamma$: transition rate of MB→MF; it is the function of the prediction reliability of the MB system based on the posterior estimation of the SPE distribution ($\chi_{MB}$) [26])

Finally, in the action selection stage, the model selects the action stochastically using the softmax rule [41]. For more details, refer Lee et al (2014).

$$P(s, a) = \frac{\exp(\tau Q(s, a))}{\sum_b \exp(\tau Q(s, b))}$$

($\tau$: degree of exploitation)

In this study, we suggested two variants of arbitration control, allowing for sub-optimality in value learning, arbitration, and action selection: one version with separate model-based and model-free learning and another version with dynamic exploitation. The former type of model assumes the different learning rates of a model-based and a model-free system. The latter class of models is based on the former model, with the further assumption that the degree of exploitation, an indicator of optimality of the RL agent's policy, is a function of the model choice probability, $P_{MB}$. We tested three different types of exploitation as follows: logistics ($\tau = c_1 + (c_2-c_1)/(1+\exp((-c_3(P_{MB}-c_4))))$, where $c_1$: lower-bound, $c_2$: upper-bound, $c_3$: steepness of the curve, $c_4$: midpoint of the function), linear ($\tau = P_{MB}{}^*c$), and weighted linear ($\tau = P_{MB}{}^*\tau_{MB} + (1-P_{MB}){}^*\tau_{MF}$). It is based on the earlier evidence suggesting that the MB and the MF system guides flexible and direct action selection, respectively [80]. In summary, we tested the following models: (1) Arbitration model proposed in (Lee et al., 2014) (Arb), (2) Arb model additionally assigning separate learning rates to model-based and model-free system (use $\alpha_{MB}$, $\alpha_{MF}$ instead of $\alpha$, $Arb_\alpha$), (3)~(5) $Arb_\alpha$ model with additional assumption that the degree of exploitation is function of the model choice probability ($Arb_{\alpha,\tau1}$, $\tau1 = f(P_{MB})$, f: logistics ($\tau = c_1 + \frac{c_2-c_1}{1+e^{(-c_3*(P_{MB}-c_4))}}$); $Arb_{\alpha,\tau2}$, $\tau2 = f(P_{MB})$, f: linear($\tau = P_{MB}{}^*c$); $Arb_{\alpha,\tau3}$, $\tau3 = f(P_{MB}, \tau_{MB}, \tau_{MF})$, f: weighted linear($\tau = P_{MB}{}^*\tau_{MB} + (1-P_{MB}){}^*\tau_{MF}$)). Note that in all cases, we set the parameters of the model in a way that is reduced to the original RL with a single exploitation parameter.

We used the Nelder-Mead simplex algorithm [81] to estimate the free parameters by minimizing the negative log-likelihood of choosing a selected-action in each state, summed over all trials for each participant. We optimized parameters 128 times with randomly generated initial seeds to prevent local optimization. For model comparison, we used a Bayes Factor [28] based on the Bayesian Information Criterion (BIC) [82]. This method penalizes each model for the number of free parameters.

### fMRI data acquisition

Functional imaging was conducted on a 3T Siemens (Magnetom) Verio scanner located in the KAIST brain imaging center (Daejeon). Forty-two axial slices were acquired with interleaved-ascending order at the resolution of 3 mm x 3 mm x 3 mm, covering the whole brain. A one-shot echo-planner imaging pulse sequence was utilized (TR = 2800 ms; TE = 30 ms; FOV = 192 mm; flip angle = 90˚). The high resolution structural image was also acquired for each subject to the resolution of 0.7 mm X 0.7 mm x 0.7 mm.

### fMRI data pre-processing

Images were processed and analyzed using the SPM12 software (Wellcome Department of Imaging Neuroscience, London, UK). The first two volumes were removed to reduce T1 equilibrium effects. The EPI images were corrected for slice timing, motion movement and spatially normalized to the standard template imaging provided by SPM software.

For the general linear model analysis (GLM), normalized images were smoothed with 6mm FWHM Gaussian Kernel and a high-pass filter (128s cut-off) was applied to remove the noise.

For the multivoxel pattern analysis (MVPA), unsmoothed EPI image data was used. De-trending and z-scoring were processed to reduce the linear trends and to match the range of the signal.

### General linear model analysis (GLM)

Subject-specific value-related signals and arbitration control signals were computed from the arbitration model, and the signals were regressed against voxel-wise signals from the EPI image set. The order of the regressors is as follows: prediction error from the model-based system (SPE) and the model-free system (RPE), reliability comparison signal which is a key variable for arbitration control (= max ($Rel_{MB}$, $Rel_{MF}$); max reliability), the chosen value of model-based system ($Q_{fwd}$), the chosen value of model-free system ($Q_{sarsa}$), the difference between chosen and unchosen integrated values ($Q_{arb}$), and the probability of selecting chosen action ($P_{chosen\ action}$). $Q_{arb}$ was defined as the chosen minus unchosen action value to reflect the view of the previous studies [26,38,83]. Subject specific design matrices were created in the following order: (R1) a regressor encoding the averaged BOLD signal at prediction error computation (the 2nd choice state and the outcome state), (R2-3) two parametric regressors encoding the prediction error (SPE, RPE), (R4) a regressor encoding the averaged BOLD signal at the value estimation (the 1st and 2nd choice states with duration as the agent's reaction time), (R5) a parametric regressor encoding the reliability comparison signal (max reliability), (R6-R8) parametric regressors encoding the values ($Q_{fwd}$, $Q_{sarsa}$, $Q_{arb}$), (R9) a parametric regressor encoding the probability of selecting chosen action ($P_{chosen\ action}$). We confirmed that there is no significant co-linearity between the 7 regressors (VIF<12 for 26 out of 28 subjects; VIF = 15 for only two subjects' $Q_{sarsa}$ regressors). The regressors were serially non-orthogonalized in the GLM analysis to prevent the effect of regressor orders in the interpretation of the results. MARSBAR software (http://marsbar.sourceforge.net) was used to extract parameter estimates from the region of interest [84].

### Whole brain analyses

To maintain the best consistency of our neural analysis, we reported neural correlates that survived after the whole-brain correction for multiple comparison at the cluster level (p<0.05 corrected) whose cluster size is more than 100 voxels, referring the previous work [26]. For right ilPFC, we ran a 10 mm small-volume correction to evaluate the effect of depression on the

arbitration control. A 10 mm radius sphere is based on the previous work which applies a 10 mm sphere criteria in the lateral PFC region [85]. The bilateral ilPFC coordinates were defined following the previous work [26]. To quantify the effect of depression on neural representations, the effect size of all peak coordinates of the activated regions are compared with the depression score [13,15]. To prevent any bias/assumptions from affecting or imposing constraints on our analysis, we compiled the list of ROIs purely based on our model-based fMRI analysis. Because our computational models were fitted to subjects' choice behavioral data, as opposed to depression-specific data, we do not expect that all of the main effects from the model-based fMRI analysis are necessarily correlated with the depression score. Accordingly, our ROI analyses employed an explorative approach. We considered all of the brain regions we identified from the model-based fMRI analysis, and reported the neural effects that passed the statistical significance test.

## Multi-voxel pattern analysis (MVPA)

The MVPA was conducted to quantify types and amounts of information encoded in the specific region of interest (ROI). The classification performance is regarded as the amount of information pertaining to the variable of interest. We chose three ROIs previously known to engage in the arbitration control process. Specifically, these areas are known to encode the reliability of both MB and MF, as well as the reliability of the strategy eventually chosen by the agent [26,30]. Masks of each brain region were functionally defined from the GLM analysis. We used the clusters whose response to the Max reliability signal survived after the whole-brain correction ($p<0.001$, uncorrected) as a mask for each ROI. We set the BOLD response time 4-6s.

A binary Support Vector Machine (SVM) classifier, an optimal supervised learning algorithm for prediction and generalization, was applied to learn voxel patterns with each ROI. For each subjects' data, the SVM was trained to best match its output to a binarized reliability-related signal (MB reliability, MF reliability, or Max reliability); the 33$^{rd}$ and 67$^{th}$ percentile threshold were used to define the two classes, 'high-value group' and 'low-value group', respectively. All voxels in the mask were used for learning. The input dimension is 196, 79, 294 for left ilPFC, right ilPFC, and FPC, respectively. The average numbers of data from each subject are 350 for MB reliability, 549 for MF reliability, and 543 for Max Reliability. Thirty-fold cross-validation was conducted for evaluation. All processes were implemented based on the Princeton Multi-Voxel Pattern Analysis toolbox [86]. Finally, a one-way ANOVA analysis was conducted to compare signal prediction accuracy between the healthy and subclinical depression groups; the two subject groups were defined by using the standard cutoff criteria of the CES-D score, 16 [31,32]. Note that the conventional cutoff criteria was suggested to identify individuals at risk for clinical depression, not for diagnosing a depressive disorder.

## Statistical analysis

We used Pearson correlation analysis to calculate the correlation between the variables and the depression score. On the other hand, Spearman correlation analysis was used when calculating the correlation between the estimated parameters of the model and the depression score, taking into account the fact that the estimated parameters were right-skewed by the lower bound value 0. One-way ANOVA was used when comparing variables between groups. T-test was used to compare the computational models.

## Supporting information

**S1 File. Supplementary information.**
(DOCX)

**S1 Data. Excel spreadsheet containing underlying numerical data for Figs 2, 3, 4, 6, 7 and 8.** Each separate sheet corresponds to the individual figure, respectively. (XLSX)

## Author Contributions

**Conceptualization:** Suyeon Heo, Sang Wan Lee.

**Data curation:** Suyeon Heo, Sang Wan Lee.

**Formal analysis:** Suyeon Heo, Yoondo Sung, Sang Wan Lee.

**Funding acquisition:** Sang Wan Lee.

**Investigation:** Suyeon Heo, Yoondo Sung, Sang Wan Lee.

**Methodology:** Suyeon Heo, Sang Wan Lee.

**Project administration:** Sang Wan Lee.

**Resources:** Sang Wan Lee.

**Software:** Suyeon Heo, Yoondo Sung, Sang Wan Lee.

**Supervision:** Sang Wan Lee.

**Validation:** Yoondo Sung, Sang Wan Lee.

**Visualization:** Suyeon Heo, Yoondo Sung, Sang Wan Lee.

**Writing – original draft:** Suyeon Heo, Sang Wan Lee.

**Writing – review & editing:** Suyeon Heo, Yoondo Sung, Sang Wan Lee.

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
