## [Decision Letter · Decision Letter 0]

17 Sep 2020

Dear Dr. Lee,

Thank you very much for submitting your manuscript "Effects of mild depression on prefrontal–striatal model-based and model-free learning" for consideration at PLOS Computational Biology.

As with all papers reviewed by the journal, your manuscript was reviewed by members of the editorial board and by three independent reviewers. In light of the reviews (below this email), we would like to invite the resubmission of a significantly-revised version that takes into account the reviewers' comments.

As the authors can see from the reports below, all referees appreciate the aim of the study and the use of the MF/MB-arbitration task in this dimensional sample. The referees' main areas of concern are (i) a lack of motivating certain analyses, (ii) the multitude of different analyses performed without clearly motivating them or correcting for multiple comparisons, (iii) a lack of details on methods and analyses. We expect the authors to address all concerns raised by the referees.

We cannot make any decision about publication until we have seen the revised manuscript and your response to the reviewers' comments. Your revised manuscript is also likely to be sent to reviewers for further evaluation.

Sincerely,

Tobias U Hauser, PhD

Associate Editor

PLOS Computational Biology

Samuel Gershman

Deputy Editor

PLOS Computational Biology

Reviewer's Responses to Questions

**Comments to the Authors:**

Reviewer #1: The authors present an well-written study investigating model-based, model-free and the arbitration between the two modes of decision making in sub-clinical depression. Specifically, they found that depression was linked to sub-optimal choices in the two-stage Markov decision task. Subsequently, they expanded a previous model of arbitration control between model-based and model-free to include separate learning rates for model-based and model-free (to allow for sub-optimality in learning values) and defining an exploitation sensitivity parameter (to allow for sub-optimality in converting learned values into choice behavior). In addition to replicating basic model-based and model-free neural correlates, the authors also examined the relationship of these representations with individual differences in depression score. There was evidence that depression was negatively linked to SPE and RPE representations, and that prediction performance of model-free reliability was higher in more depressive participants, suggesting that depressive participants' arbitration of action control are over-sensitive to model-free updates. Finally, exploitation for model-based RL was also negatively associated with depression. Together, these findings highlight various reinforcement learning deficits in depression in model-free, model-based as well as in the arbitration of the two systems.

The Abstract and Summary were concise and accurate, and the main text was easily readable. I am generally impressed by the rigorous report and data presented. The analyses and discussion were also very comprehensive. However, there are several points below which I hope the authors can address:

- I find the description of the task in both Fig 1 and in the Methods slightly unclear, which may have contributed to my confusion about the task paradigm. The authors state that there are four main sessions with 80 trials: Is this 80 trials per session or 80 trials overall? Where the transition probabilities are manifested is also unclear. There are two state-transition probabilities (0.5, 0.5) and (0.9, 0.1), which are later mentioned to be interleaved with the uncertainty condition in a factorial design. Does this mean that for a trial with (0.9, 0.1) probability, the 0.9 probability applies to two transitions, for one of both the left and right S1 choice each, and vice-versa for the 0.1 probability? Similarly for the stage-two to outcome transitions, do these also follow the (0.5, 0.5) and (0.9, 0.1) probabilities of the earlier the state transition? This should be depicted more explicitly Figure 1a as well.

- I would suggest that the authors include a signpost before the neural results that the fMRI data was examined with a subset (n=28) of the total behavioral group in the main text.

- Was there any power analysis conducted for this study/How was the sample size determined?

- The authors mention that the learning of state-action value is unrelated to depression, however I struggle to find these results in the Results section.

- It was highlighted that these RL deficits observed here are likely attributable to depressive, rather than anxiety, symptoms, which is a concern due to their high co-morbidity. Has there been specific studies of anxiety showing these similar deficits in RL, and did those studies control for depression?

- Could the authors elaborate on this statement on pg 17, line 4-6. "First, simply evaluating the two separate hypotheses contradicts the prevailing view that the brain circuitry guiding goal-directed and habitual behavior interact with each other."? I don't understand how current evidence goes against the view.

- The participant group was split into two groups for the MVPA analysis, a "depressive" group and a "control" group. This classification is sudden given that the participants are not clinically diagnosed. In the methods it was mentioned that these groups are formed based on the clinical cut-off with the CES-D - this should be also referred earlier in the results. Another point is that there seems to be quite a high number (nearly half?) who cross the threshold for a depression diagnosis with this metric, especially from a recruitment that explicitly excluded psychiatric diagnoses. Is this common/expected in the sample recruited?

Minor comments:

- Is age/gender a co-variate in any of these analyses?

- Pg 9, line 2, P(MF) is not defined (1 - P(MB)) ?

- Table E: GAD score, in the legend it says n = 23, but in the table it says n = 51?

- Table D results could be represented in a landscape format, to compare Lee et al vs the current results side by side.

Reviewer #2: The authors use a multistage choice task, fMRI, and computational modeling to examine how mild depression influence participants’ mediation between model-based and model free decision-making. The authors show that depression severity is negatively correlated with depressive severity, such that those individuals with lower levels of depression exhibit better performance (i.e., higher earnings through the course of the experiment). The authors fit the behavioral data using a modified reinforcement learning that accounts for model-based and model free choice, and the arbitration between the two; they use this modeling along with neuroimaging data to show that mild depression is associated with attenuated state and reward prediction error representation in the insula and caudate.

This is a potentially interesting paper that is ambitious in its integration of psychiatric classification of participants, behavioral modeling, and model-based fMRI. However, I am not convinced that the authors’ main findings are statistically valid and that the associated behavioral modeling and neuroimaing are combined in a scientifically coherent way. I’ve outlined my concerns below.

Behavior:

-Figure 1 shows correlations between accumulated reward, proportion of optimal choices, choice consistency, and depression score. However the authors did not justify their use of the Center for Epidemiologic Studies Depression (CES-D) questionnaires. How and why was the scale chosen? Were other scales used? This scale has been shown to not align very well with DSM diagnosis of depression.

-Why do the authors only relate model free behavior to depression measures (Fig 1)? It seems that if their computational models are truly sensitive to model based/model free arbitration their model parameters should be related to CES-D measures of depression. Was this not the case?

-Regarding behavioral modeling I’m curious about the extent of correlation between tau_mb and tau_mf and other RL model parameters. How behaviorally dissociable are these model parameters? This is important because it will impact the model-based neuroimaging that relies on these model parameters.

-Overall, it seems strange that the authors’ computational model of arbitration does not covary with depression symptoms, as in the average behavior. This make me wonder if model-based/ model free arbitration is really influenced by depression, as the authors suggest. This is the main focus of the paper, so this point really needs to be flushed out.

Imaging

-It is great that the authors replicate the neuroimaging main effects from their 2014 paper. These correlations appear to be very strong.

-The correlations between depression scores and neural effects (figure 5) seem to be very weak and not well motivated. The authors obtain strong bi-lateral insula and dlPFC activity in the main effect, but focus on left insula in the correlation with depression. In the RPE contrast they get strong vmPFC and putamen activity but individually look at left and right putamen correlations with depression score. More needs to be done to justify these ROI analyses.

-I would also be interested to know if depression score correlates with Q-value or Pchosen. This is important information. What brain activity best describes the depression scores? This isn’t presented in the paper, so it is hard to understand the strength of neural results in the context of depression.

Reviewer #3: In their manuscript “Effects of mild depression on prefrontal–striatal model-based and model-free learning”, Heo et al. used behavioural testing, computational modelling and fMRI to examine MB and MF reinforcement learning in humans. The authors found that mild depressive symptoms are associated with reduced neural prediction error signals and impaired arbitration between MB and MF learning. Lastly, the authors also show that mild depression is related to reduced exploitation of rewarding options.

1) The authors mainly present correlational results between CES-D scores and behaviour/neural data. In the figures, however, and on p16 in the results section, the authors also refer to one group as “healthy”, and the other as “depressive”. I might have missed this, but I didn’t find a proper explanation for how these groups were defined in the methods section. Based on a particular cut-off score for symptoms? Based on a median split? Given that, as far as I understand, all subjects are supposed to be non-diagnosed with clinical depression by a psychiatrist or specific clinical standards, this seems highly problematic to me. Overall, I do not think that authors can say mild depression, given that subjects are not diagnosed with depression. It seems to me that some subjects seem to have subclinical depression, i.e., depressive symptoms but not necessarily meeting criteria for depressive disorder. Also, it is not appropriate to refer to subjects as having “early” depression. This study is cross-sectional, and we do not know anything about the trajectory of symptoms of the subjects. The authors need to be clearer about their procedure, and in turn, define their research questions appropriately.

2) The authors present three correlations of model-free behavioural data and depression scores (Fig. 2). What is the rational for presenting these correlations only? I am sure, there are many other behavioural model-free variables? Would be good to know whether the authors had specific hypotheses? Also, I am not sure if I understand the reasoning for using Pearson and Spearman. Are the depression scores normally distributed?

3) I am wondering whether the authors could relate the modelling results more to the model-free results they are showing in Fig. 2? As is, it appears subjects with higher depression scores are performing poorly in the task. Can the authors unpack that, potentially explain it with the results from the modelling, i.e., differences in learning? There is no relationship of results from model-free and model-based analyses mentioned as far as I can tell.

4) Related to 3): subjects with higher depression scores perform worse. Thus, I am wondering whether the authors have collected any other cognitive scores for subjects to further unpack this result.

5) Related to 2): would be important to know what the relationship between the three task measures from Fig.2 is, e.g., how choice consistency is related to task performance, so as to understand the link between these correlations, and the behavioural performance in the task in general.

6) In Fig. 3, the authors nicely explain the different models tested. However, for the understanding of the modelling section, it would be great if the authors could provide more detail in the methods, e.g., the fact that they use two distinct learning rates, αMB & αMF. I’d suggest the authors provide the equations for the different models in more detail.

7) Why are the authors specifically thresholding to a cluster size of 100 voxels (p29)?

8) “For some areas reported from previous studies, we ran a 10 mm small-volume correction.” Given that the authors present a multitude of imaging results, which is not easy to follow when reading the manuscript, the above sentence is not appropriate. The authors need to be clear about i) which regions they looked at for which contrasts, ii) why they have chosen these regions, based on the canon of knowledge in the literature, iii) how many tests they ran for how many different regions (multiple comparison correction!), iv) why they have chosen 10mm (because for some, small brain regions, this seems quite a large threshold), v) how these regions were defined, using pre-defined anatomical masks or previous imaging studies as reference, and vi) whether they used a sphere/mask etc.? There is much more information needed here for the reader to understand this.

9) The authors report that 65 subjects did the task, of which 28 were scanned. Did the fMRI subjects perform two sessions? Or only one (in the scanner)? Were the timings etc. identical between behavioural only and imaging?

10) How long was the task (in minutes, volumes for fMRI)? What is the exact timing of the trials? There is a lot of methodological information missing as far as I am concerned.

11) There is a lot of causality implied in the wording. That is, depression (which is critical, given as mentioned in 1), subjects are not even diagnosed with depression!) “disrupts” (abstract) or “driven by” (p16). “First, our neural results explain why…” (p21). I’d suggest the authors be more cautious here, because the find correlational relations between sub-clinical symptom data from questionnaires and task data, but it is not appropriate to infer causality from this data.

12) In the discussion section, the authors suddenly mention anxiety symptoms. This is not mentioned before, also not in the methods section. Thus, I am unclear what to make of this. Is there a relationship between anxiety and depressive symptoms in this group of subjects (which you’d expect)? This data needs to be presented to properly understand these results. Then, in the limitations section, the authors mention that there is a lack of other measures, such as anxiety. This is very confusing.

13) What does “peak-level corrected” refer to in the fMRI results section? Is this p<0.05, FWE whole brain? This needs to be clearly stated at some point in the manuscript, e.g., in the methods.

14) “It would be possible that this model can be used to predict the severity of mild depression as this model is fitted with individual behavioral data.” (p17). This seems an overly strong statement to me. The authors need to explain this, and why they think this is the case, given the data at hand.

15) “However, as other possibilities still remain to be explored, e.g., intact RPE representation in MDD [46]” (p19). May be worth discussing that Rutledge et al. found this in a non-learning task, which is different to the one the authors used here.

16) What does it mean that the two subjects that were excluded performed “below the chance-level” (p23)?

17) “pre-learning” and “pre-training” (p23), should this not be the same?

18) “p<0.001, uncorrelated” should be “uncorrected” (p30).

**Have all data underlying the figures and results presented in the manuscript been provided?**

Reviewer #1: Yes

Reviewer #2: **No: **I did not see source data associated with the figures.

Reviewer #3: Yes

PLOS authors have the option to publish the peer review history of their article (what does this mean?). If published, this will include your full peer review and any attached files.

Reviewer #1: No

Reviewer #2: No

Reviewer #3: **Yes: **Jochen Michely
---

## [Decision Letter · Decision Letter 1]

5 Jan 2021

Dear Dr. Lee,

Thank you very much for submitting your manuscript "Effects of subclinical depression on prefrontal–striatal model-based and model-free learning" for consideration at PLOS Computational Biology. As with all papers reviewed by the journal, your manuscript was reviewed by members of the editorial board and by several independent reviewers. The reviewers appreciated the attention to an important topic. Based on the reviews, we are likely to accept this manuscript for publication, providing that you modify the manuscript according to the review recommendations.

Specifically, reviewer 3 had a few more clarification questions that we kindly ask you to address in a further revision. Please also see the suggestions from reviewer 1, who made some suggestions to further improve the readability of this paper.

Sincerely,

Tobias U Hauser, PhD

Associate Editor

PLOS Computational Biology

Samuel Gershman

Deputy Editor

PLOS Computational Biology

[LINK]

Reviewer's Responses to Questions

**Comments to the Authors:**

Reviewer #1: I thank the authors for the comprehensive response to my comments. I appreciate the more nuanced label of 'subclinical depression'. The changes to the task figure are helpful and the revision of the text with signposting has made the results a lot clearer and easier to follow.

A few things I would recommend:

1. Author should use numeric numbers (e.g. N = 63, instead of 'sixty-nine') for the number of participants (both in results and methods) for better readability.

2. The discussion would read better if the last paragraph is a concise conclusion, instead of ending abruptly after a limitations paragraph.

Overall, I believe the authors have sufficiently addressed my concerns.

Reviewer #2: The authors have done a good job responding to my comments. I have no further comments.

Reviewer #3: I congratulate the authors on their job in clarifying most of the issues raised. I am of the opinion that the paper is much improved. However, I am still missing some information in their responses.

1) re. 3.1: The authors now explain how they split up the sample in two groups, but as far as I can tell only for the fMRI sample (n=28). I am wondering, however, how they split up the groups in the overall sample (n=63)?

2) re. 3.3: “subjects with a lower depression score more likely to use…”? Shouldn’t that be "higher" depression score?

3) 3.5: The response given by the authors is not sufficient in my view. I was asking about how the three main model-free measures are related, i.e., what is the relationship between overall learning, model-based learning, and model-free learning, across the entire sample? This is not explained in 3.3, as stated by the authors.

4) I advise the authors to show the changes they made in the manuscript also in the response letter to reviewers. As it is, I cannot infer the changes that have been made by the authors in response to 3.6, 3.8, 3.11, which makes it impossible for me to review the changes.

**Have all data underlying the figures and results presented in the manuscript been provided?**

Reviewer #1: Yes

Reviewer #2: Yes

Reviewer #3: Yes

PLOS authors have the option to publish the peer review history of their article (what does this mean?). If published, this will include your full peer review and any attached files.

Reviewer #1: No

Reviewer #2: No

Reviewer #3: No
---

## [Decision Letter · Decision Letter 2]

26 Apr 2021

Dear Dr. Lee,

We are pleased to inform you that your manuscript 'Effects of subclinical depression on prefrontal–striatal model-based and model-free learning' has been provisionally accepted for publication in PLOS Computational Biology.

Best regards,

Tobias U Hauser, PhD

Associate Editor

PLOS Computational Biology

Samuel Gershman

Deputy Editor

PLOS Computational Biology

Reviewer's Responses to Questions

**Comments to the Authors:**

Reviewer #3: The authors have clarified all remaining issues. I congratulate the authors on their work.

**Have the authors made all data and (if applicable) computational code underlying the findings in their manuscript fully available?**

Reviewer #3: None

PLOS authors have the option to publish the peer review history of their article (what does this mean?). If published, this will include your full peer review and any attached files.

Reviewer #3: No

---

## [Editor Report · Acceptance letter]

11 May 2021

PCOMPBIOL-D-20-01407R2 

Effects of subclinical depression on prefrontal–striatal model-based and model-free learning

Dear Dr Lee,

I am pleased to inform you that your manuscript has been formally accepted for publication in PLOS Computational Biology. Your manuscript is now with our production department and you will be notified of the publication date in due course.

With kind regards,

Katalin Szabo
